# Stakeholder acceptability of the risk reduction behavioural model [RRBM] as an alternative model for adolescent HIV risk reduction and sexual behavior change in Northern Malawi

Marisen Mwale[1,2]*, Adamson S. Muula[3,4]

1 Department of Public Health, School of Public Health and Family Medicine, University of Malawi College of Medicine, Blantyre, Malawi, 2 Department of Education Foundations, Mzuzu University, Mzuzu, Malawi, 3 Department of Public Health, School of Public Health and Family Medicine, Kamuzu University of Health Sciences, Blantyre, Malawi, 4 Africa Center of Excellence in Public Health and Herbal Medicine, Kamuzu University of Health Sciences, Blantyre, Malawi

* marisen.mwale@yahoo.co.uk

**Data Availability Statement:** All relevant data is made available here DOI: 10.13140/RG.2.2.33809. 97124.

## Abstract

We sought to assess stakeholder acceptability of a risk reduction behavioural model [RRBM] designed for adolescent HIV risk reduction and whose efficacy we tested in selected schools in Northern Malawi. We used qualitative procedures in sampling, data collection and data analysis. Our data collection instrument was the semi-structured interview and we applied thematic content analysis to establish stakeholder evaluations of the RRBM model. The study population included 10 experts working within key organizations and teachers from two schools. The organizations were sampled as providers, implementers and designers of interventions while schools were sampled as providers and consumers of interventions. Individual study participants were recruited purposively through snowball sampling. Results showed consensus among participants on the acceptability, potential for scale up and likelihood of model sustainability if implemented. In essence areas to consider improving and modifying included: focus on the rural girl child and inclusion of an economic empowerment component to target the underlying root causes of HIV risk taking behavior. Stakeholders also recommended intervention extension to out of school adolescent groups as well as involvement of traditional leaders. Involvement of parents and religious leaders in intervention scale up was also highlighted. The study serves as a benchmark for stakeholder involvement in model and intervention evaluation and as a link between researchers and project implementers, designers as well as policy makers to bridge the research to policy and practice gap.

## Introduction

Sustainable development goal number 3 targets HIV incidence reductions among young people and the general population by 2030 [1]. Malawi with HIV prevalence of 8.8% in the 15–49 years age category is one of the sub-Saharan African countries with a high prevalence and

**Funding:** I would like to notify that the study was funded as part of institutional Staff Development at my work station –Mzuzu University who provided partial funding covering tuition and part of the research.

**Competing interests:** The authors have declared that no competing interests exist.

generalized epidemic [2, 3]. Adolescents and young people are one of the key populations considered highly vulnerable to new HIV infections [3]. The national HIV prevalence in the 12–24 years age category is estimated at 5.9% an increase from 3.6% in 2010 [2, 4]. In 2012 alone, there were 6 700 new HIV infections in the 15–19 years category translating to 18 infections on a daily basis [1]. In 2013 there were more than 34 000 infections among young people and of the 36 000 overall new infections in 2016, more than half were in the 12–24 years age group [2, 3].

We designed an HIV intervention model–the risk reduction behavioural model (RRBM) for adolescents with the hope of bridging the intervention vis-à-vis behaviour gap. Previous studies and meta-analysis indicate changes in knowledge but limited behavioural outcomes after interventions [5–11]. Interventions have mainly registered knowledge changes that do not translate to expected behavioural outcomes [12–19]. HIV risk reduction measures according to Hearst and Chen [20] include: abstinence, late sexual debut, consistent condom use, faithfulness to one partner, non-involvement in multiple and concurrent partnerships, non-involvement in culturally related sex, non-involvement in sex under drug, alcohol and substance influence. For our initial study, specificity situation analysis [21] to identify correlates for the Northern region informed model design and made it culturally and contextually tailored.

We tested for model efficacy through a quasi experiment [16] and results were significant on diverse HIV risk reduction and sexual behavioural change outcomes. We therefore conducted a key stakeholder acceptability study within the same research site where the efficacy study was rolled out. The main purpose of the stakeholder acceptability study was to determine perceptions among stakeholders–that is designers, implementers and providers on the acceptability [22] of our RRBM model. Objectives were threefold: first, to assess key stakeholder acceptability of the RRBM model. Second, to determine what modifications need to be done for the RRBM model. And third, to determine the potential for scale up and sustainability of the RRBM model if extrapolated to the entire population of adolescents in Malawi. This paper presents the findings for the stakeholder acceptability study.

## Methods and materials

We used a descriptive design applying the qualitative approach in sampling, instrumentation and data analysis. Stakeholders involved in providing, designing, implementing and consuming adolescent sexual and reproductive health (ASRH) programmes and interventions in Northern Malawi constituted the research population. The study sites were the same as for the formative situation analysis phase [21] and the confirmatory intervention phase [16]—i.e. Mzuzu City, Mzimba and Nkhata Bay districts. Those in the focal areas of HIV prevention, sexual reproductive health and sexual behavioural change were prioritized. The sample included experts from key organizations and teachers working with adolescents in ASRH programmes that include HIV prevention as a component.

We sampled 10 experts and two teachers for involvement into the study. The snowball sampling procedure was applied in identifying study participants within organizations through project managers. Some of the organizations were already targeted in the first phase of the entire research project [21] and included: Plan Malawi, World Vision Malawi, Population Services International (PSI), Campaign for Girls Education (CAMFED), National AIDS Commission (NAC), Livingstonia Synod AIDS Programme (LISAP), Girl Empowerment Network (GINET), Ekwendeni Hospital HIV and AIDS programme, and the District Health Office. Education stakeholders included teachers and head teachers sampled from Katoto and Luwinga Secondary schools. Organizations (non-governmental-NGOs) were sampled as

providers, implementers, designers of interventions and programmes while schools were sampled as providers and consumers of interventions.

To be included, an organization was supposed to be:

- Focusing on and working in impact areas of adolescent HIV prevention and sexual reproductive health,

- Implementing some form of BCI in projects and/or programmes

- Within the study sites and setting of the research project.

Considering that our study was qualitative and also that few organizations are directly involved in adolescent HIV prevention and sexual reproductive health, we may have been limited in options and hence the purposive sampling approaches.

In data collection we used semi-structured interviews. A week before the interviews, experts were furnished with the detailed RRBM model as summarized in the methods subsection, 'The Risk reduction behavioural model'. The full details of the model are included in S1 File. Other supporting documents included publications on the situation analysis and intervention phases of the study. Provision of the model and study publications was done to allow for evaluation, assessment and consultation within organizations. An interview followed a week later informed by the interview guide (S2 File). Interview items focused on acceptability, potential for model scale-up and on areas for improvement or perhaps modification. Interview guiding questions were piloted in two non-participating organizations before data collection to ensure content reliability of items.

The data collection process at each of the 10 organizations specifically involved the PI identify, brief, and have an expert at a respective organization or institution go through consent forms, the model and supporting documentation a week prior to interviews. The process therefore ensured that the right expert or informant was identified, furnished with the model, arranged an appointment with the PI for an interview a week later. The week grace accorded a period to appraise the model, supporting documents and assent to participate or not before the actual interviews. During the period scheduled for interviews, the PI would first call the expert or informant by phone to confirm their availability. This was despite booking an appointment earlier in the previous week as most experts in organizations as first line managers have busy schedules that could change their programmes any time. After confirmation of availability, the PI would then go to the scheduled organization, Plan Malawi for instance to interview the expert.

At each of the organizations, interviews were conducted in the organizational board room while some experts or informants preferred their office. Before the session, the participant would be de-briefed on the objectives of the study, that participation was voluntary and that if for any reasons they felt obliged not to participate they had freedom to do so. None of the experts who initially assented to participate withdrew from the study. Interviews were face-to-face for purposes of clarity and the PI conducted them personally. Each interview session was scheduled for 30–40 minutes although some sessions ended up shorter or a little longer than planned depending on the responses and ensuing probing. The PI strictly adhered to the interview guide even when some participants were inclined to digress or be overzealous to furnish more information.

After data collection, the recorded interviews were checked for consistency before transcription and coding for data analysis. The PI with assistance from a qualitative data analyst at our University and verification from the co-investigator analysed the data. We analysed data through thematic content analysis. More specifically; audio-recorded semi-structured interview data was transcribed, translated, text bracketed, gleaned, winnowed, categorized using

constant comparative procedure, thematized and finally theorized. Themes and sub-themes were coded, categorized and tabulated in line with study objectives. Altogether five main themes were identified. The themes in line with respective objectives included:

**Objective 1** –Identifying whether model is acceptable or not.

Theme 1- acceptability of model.

**Objective 2** –Exploring areas needing modification.

Theme 2 –factors to facilitate implementation effectiveness.

Theme 3 –factors associated with youth priorities and needs.

Theme 4 –structural and contextual factors.

**Objective 3** –Exploring potential for scale up and sustainability.

Theme 5- model potential for scale up and sustainability. This whole process is consistent with and follows guidelines for qualitative data analysis by Merriam [23] and Seidman [24]. No software was used in analysis. Consensus was reached after the three of us went through and agreed on tabulated themes and categories and after verification by participants. The digitally recorded type copies of interviews were kept under password protection for confidentiality purposes and have been deposited with researchgate for public access by researchers and other interested stakeholders [25] (DOI: 10.13140/RG.2.2.33809.97124).

Ethics approval for the study was granted by the University of Malawi College of Medicine Research and Ethics Committee (COMREC) on 12 July 2019 [COMREC # P.04/19/2652]. We conducted our study in full cognizance of ethical principles and statutes guiding research with human participants such as those specified by the Helsinki Declaration and Charter of Fundamental Rights of the European Union. We took account of stipulated guiding research principles as: anonymity, informed consent, confidentiality and other such data management and protection issues in methods design, data collection, and data analysis as well as interpretation. Specifically, all participants who assented to participate signed consent forms. Objectives of the study were explained a priori and participants were assured of the confidentiality of their input. No any form of personal identifiers or names were used during interviews to ensure anonymity and confidentiality of input. The participant compensation for time accorded was $ 10. The following section within methods summarizes the RRBM as presented to experts as part of the stakeholder acceptability and model evaluation process.

## The risk reduction behavioural model

The Risk Reduction Behavioral Model on which our intervention was grounded (S1 File) as presented to stakeholders for evaluation is an integration of: social-cultural theory [26, 27], socio- learning theory [28], and the theory of reasoned action [29]. The intervention included an information, efficacy [self and interpersonal /social] and skills [coping and practical] building package. This package was modular and complemented with entertainment as recommended by participants in the first situation analysis phase of the study [21]. In the social infotainment package, study participants were involved in formal HIV risk reduction knowledge, skills and efficacy building sessions that also included: drama, song, music, film and role—playing. In each school five sessions spread over a three months intervention period were implemented. Fig 1 below illustrates the risk reduction behavioural model.

Messages and skills were transferred through the participatory approaches in which the more knowledgeable facilitators played the role of mentors; instilling, cuing and modeling expected behaviours in line with the scaffolding and modeling constructs of socio-cultural and social learning theories. We distributed pamphlets and booklets containing content on modes of HIV transmission, strategies for avoiding or reducing HIV contraction, how to minimize AIDS stigma and discrimination and on general sexual reproductive health. The control group

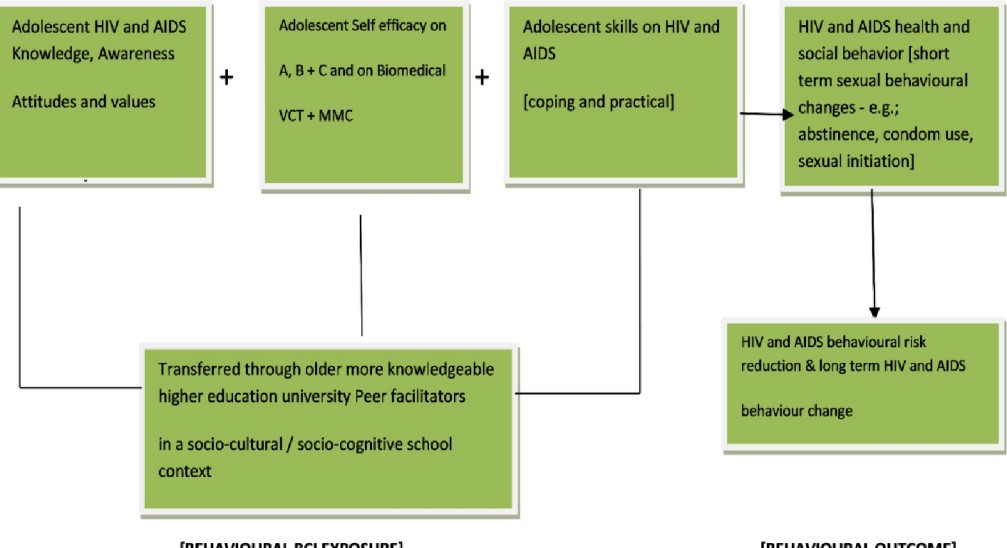

**Fig 1. The risk reduction behavioural model.**

was exposed to a Health promotion and education package based more on the standard-as-usual Life skills BCI regimen already being implemented in all schools. Instead of HIV risk reduction knowledge and skills, they were exposed to malaria and tuberculosis risk reduction and prevention skills. Apart from that they were also exposed to generic Life skills incorporating decision making, assertiveness and self-esteem building. We hypothesized that the intervention would result in positive sexual and reproductive health behaviour. The paper presenting the detailed results for the interventional study published in the Journal of HIV and Social Services [16] was submitted to the experts for evaluation as part of the risk reduction model stakeholder acceptability process.

## Results

Informed by the three study objectives: first to assess stakeholder acceptability of the RRBM model, second to determine what modifications need to be done, and third to determine potential for scale up and sustainability; we identified five themes from the semi-structured interviews with participating stakeholders. We present results as they emerged from participants in line with the three objectives and associated themes in this section.

### Objective 1—Assessing stakeholder acceptability of the model

**Theme 1: Whether model is acceptable or not.** There was consensus among stakeholders on the acceptability, potential for scale up and potential odds of sustainability of the RRBM model if implemented. Some of the reasons cited as potentially ensuring acceptability included that: the model is robust, innovative, practical, cost-effective and pragmatic. Stakeholders pointed to innovativeness in design and potential likelihood to be owned by primary adolescent beneficiaries if scaled up to the adolescent population in Malawi. Further, there was mention of the model resonating with adolescent and youth expectations for HIV risk reduction, prevention and skills building. Table 1 below illustrates participant input on acceptability of RRBM model–objective 1 / theme 1.

**Table 1. Participant input on acceptability of RRBM model [Objective 1/Theme 1].**

| Objective 1 | Theme 1 | Participant input as quoted |
|---|---|---|
| Acceptability of RRBM model | Theme: whether model is acceptable or not | • model is robust, practical, pragmatic and acceptable<br>• compatible with impact area rural contexts<br>• model acceptable and a departure from the business as usual mentality<br>• innovative design and implementation<br>• cost effective considering that there may be no need for a lot of external resources<br>• potential to be owned by beneficiaries<br>• it is a move towards sustainability<br>• top down design commendable as adolescent primary beneficiaries co-developed model<br>• addresses issues affecting young people in communities |

## Objective 2—Areas needing modification

**Theme 2: Factors to facilitate implementation effectiveness.** In line with objective 2 on areas needing modification we isolated three themes from semi-structured interviews with stakeholders. The first theme being, factors associated with implementation effectiveness. Among the suggested areas were the need to address issues of teacher conservatism that often hamper self expression and open mindedness in adolescent students on issues of sexuality and reproductive health. Participants were also of the view that the intervention needs to be extended to communities and to target out of school adolescents as well. In line with the recommendation one participant opined and we quote:

'*Out of school adolescents are often at heightened risk but are mostly neglected as most interventions are school-based, this high risk group therefore needs to be targeted as well.*' [stakeholder participant in semi-structured interview]

There was also a pointer in proposing areas for strengthening the intervention in terms of implementation effectiveness to target the rural girl child who according to participant observations is often missed by interventions but bears the brunt of structural pitfalls such as poverty, gender disparities and culture. Apart from the rural girl child, some participants mentioned the need to further consider adolescents with disabilities and special needs who are disadvantaged by being embraced into one-fits-all strategies although they may have slightly differing dynamics. Table 2 below illustrates participant input on areas needing modification–objective 2 / theme 2.

**Table 2. Participant input on areas needing modification [Objective 2/ Theme 2].**

| Objective 2 | Theme 2 | Participant input as quoted |
|---|---|---|
| Areas needing modification | Theme: factors to facilitate implementation effectiveness | • implementation in schools needs to address issues of teacher conservatism that hampers self expression in students on issues of sexuality<br>• the intervention needs to be extended to communities and target out of school adolescents as well<br>• in implementation there is need to incorporate a structural economic youth empowerment component and perhaps vocational training component<br>• deliberate targeting of girls especially from the rural populations who lack basic needs<br>• deliberate targeting also of students with disabilities and special needs who are often neglected by one-fits-all approaches<br>• co-ordination between diverse stakeholders to avoid contradictions necessary |

## Objective 2—Areas needing modification

**Theme 3: Factors associated with youth priorities and needs.** The other theme isolated from objective 2 was that of youth priorities and needs. Participants were of the view that in implementation there was need to revisit the Ministry of Education (MoE) position on contraceptives especially on condom distribution in schools. One participant is quoted as having opined:

'*Although head teachers accept that school dropouts among adolescent girls are high due to teenage pregnancies, they may not want to acknowledge that condom distribution in schools would reduce the prevalence of teenage pregnancies.*' [stakeholder participant in semi-structured interview]

Mention was also made of the need to incorporate in design and implementation modern trends perhaps associated with globalization and modernism driving youth subcultures and dynamics. Youthful trends such as social media, modern technology, pop and hip-hop culture as well as other trends like theatre, the discothèque, and movies were highlighted respectively. Table 3 below illustrates participant input on areas needing modification–objective 2 / theme 3.

## Objective 2—Areas needing modification

**Theme 4: Structural and contextual factors.** Theme 4 isolated from participant input on areas needing modification was associated with the need to focus on structural and contextual factors. Participants stressed the need to engage traditional leaders, parents, religious leaders and other gate keepers in implementation and scale up. One participant opined and we quote:

'*Traditional leaders as custodians of culture should be involved in implementing the intervention as well as parents and religious leaders who are agents of adolescent and child socialization.*' [stakeholder participant in semi-structured interview]

Involvement of these agents of socialization such as the family, the church, and cultural players in intervention implementation was justified by the fact that lack of involvement of these critical agents often impedes successful achievement of outcomes. In line with that, stakeholders also recommended co-ordination between various players who implement

**Table 3. Participant input on areas needing modification [Objective 2 / Theme 3].**

| Objective 2 | Theme 3 | Participant input as quoted |
|---|---|---|
| Areas needing modification | Theme: Youth priorities and needs | • the school through the MoE needs to revisit their position on contraceptives like the condom–Head Teachers not willing to acknowledge dropouts linked to teenage pregnancies can be mitigated through provision of contraceptives in schools<br>• focus in implementation should be on already existing youth structures handling youth emerging challenges in communities and perhaps lacking youth empowerment and capacity building<br>• there is need within the nation for local youth platforms and forums for youth to express themselves–we need not look much to international forums like the Barrack Obama Foundation or Nelson Mandela Foundation for inspiration, we need our own youth platforms<br>• there is in implementation need to focus on modern trends driving youth dynamics like the social media, theatre, movies, music, disco, pop-culture, hip-hop and other global trends resulting from globalization |

**Table 4. Participant input on areas needing modification [Objective 2/ Theme 4].**

| Objective 2 | Theme 4 | Participant input as quoted |
|---|---|---|
| Areas needing modification | Theme: structural and contextual factors | • traditional leaders need to be involved in interventions for adolescents and youth as custodians of culture and as likely barrier to knowledge translation,<br>• parents as primary agents of child and adolescent socialization, religious leaders and other gate keepers also need to be involved<br>• focus should be on traditions and cultures affecting adolescent HIV risk exposure that need modification<br>• entry point criteria, should address the knowledge transfer gaps emanating from culture, religion, gender disparities and other structural bottlenecks militating adolescent behaviour |

interventions. They mentioned duplication of efforts and perhaps contradictions emanating forth as often confusing adolescent beneficiaries. One stakeholder was of the view that:

*'Harmonization and co-ordination of efforts by various players in implementation is necessary to avoid contradictions and duplication of efforts'*[stakeholder participant in semi-structured interview]

There was also mention of the need for implementation to commence from primary school. This came within the observation that training and skills building at a tender age might ensure realistic outcomes of the intervention. Table 4 below illustrates participant input- objective 2/ theme 4.

## Objective 3—Potential for scale up and sustainability

**Theme 5: Whether model has potential for scale up and sustainability?.** The last objective was to determine potential for model scale up and sustainability. Theme 5 isolated from the objective was associated with scale up and sustainability consensus among diverse stakeholders centering on model robustness and likelihood to be sustainable if scaled up. Stakeholders viewed the model as context and population specific and hence more likely to be self-sustaining. One participant was of the view and we quote:

*'The fact that the model sets mindfulness on socio-cultural factors and context in which behaviour unveils gives it a high probability of success, actually it is a move toward sustainability.'* [stakeholder participant in semi-structured interview]

Table 5 below illustrates participant input on potential for model scale up and sustainability —objective 3/ theme 5.

## Discussion

The study findings are consistent with those in similar studies across sub-Saharan Africa and elsewhere [8, 18–22] as well as with systematic reviews and meta-analysis [5, 10, 15, 17, 28–47]. The current trend in intervention implementation for instance is a shift toward combination options and structural interventions in line with recommendations by researchers and practitioners to incorporate empowerment components in intervention designs [11, 13, 14, 48–50].

There was an observation that school-based interventions might lack effectiveness due to factors linked to overall implementation approach. Teachers were cited as needing capacity building to reduce conservatism that many have toward intervention packages or some components of interventions. School based interventions especially those that incorporate the

**Table 5.  Participant input on model potential for scale up and sustainability [Objective 3/ Theme 5].**

| Objective 3 | Theme 5 | Participant input as quoted |
|---|---|---|
| Potential for scale-up | Theme: Whether model has potential for scale up and sustainability? | • model robust and has high potential for scale up and sustainability<br>• contextually driven and has high potential for adaptation into current adolescent programmes<br>• robust for adolescent skills transfers and practical sustainability of outcomes and impact<br>• potential for scale up to entire adolescent population in Malawi and could be self-sustaining due to mindfulness on socio-cultural dynamics<br>• for sustainability, in stakeholder consensus building for scale up and implementation young people ought to be involved<br>• to guarantee sustainability, during scale up there is also need to begin from primary school because commencing training and skills building at a tender age will ensure sustained impact |

condom have been noted to lack effectiveness due to teacher attitude, ambivalence and disdain [35, 51–63]. Stakeholders also cited with respect to contraception, the need for co-ordination at the status quo between the Ministry of Health (MoH) and Ministry of Education (MoE).

It was observed that while the former through guidelines in ASRH encourages and promotes the use of contraceptives among youth including the condom; the MoE on the contrary through its policies on education discourages condom promotion and programming in schools. Muchabaiwa and Mbonigaba [60] in a study conducted in Zimbabwe note that such lack of co-ordination in government policy on contraception might result in some components in interventions playing a deleterious role, with the likelihood of cancelling the effect of the overall combination. To ensure sustainability of the intervention on scale up, stakeholders also cited the need from design, through consensus building and implementation to involve traditional leaders, parents, religious leaders and other social gate keepers. Traditional leaders were recommended as custodians of culture and potential barriers or facilitators to knowledge and skills translation. Parents were recommended for involvement as primary agents of child socialization and religious leaders as potential barriers or facilitators on components like contraception due to moralistic norms.

These recommendations are in line with ecological and socio-cultural models [26–28] highlighting the need for multi-level approaches that involve all players at all levels of social or ecological systems in cognizance of social norms, values and other traditional relics in implementing interventions. The observation is also consistent with Mannell et al. [64] Michielsen [65] and Maticka-Tyndale and the HP4RY Team [66] recommendation for interventions to shift from theoretical frameworks focusing on individual risk factors or determinants of sexual behaviour to those focusing on socio-ecological domain. Focusing on individual determinants of sexual behaviour at the expense of socio-cultural or other ecological correlates in communal societies of sub-Saharan Africa has been empirically demonstrated to yield sub-optimal outcomes and to militate against intervention effectiveness [57, 58, 60, 61, 64, 65, 67, 68].

There was also mention of the need to incorporate modern trends currently driving youth subcultures and sexual dynamics such as: the social media, modern pop or hip-hop culture, music, movies, the film, the discothèque and perhaps theatre and such other trends emanating from modernism and globalization. Youth in the 21st century cannot be compared to their predecessors in the 70s for instance. Modern technology including the social media and the internet has greatly transformed youth sexual dynamics and youth are globally closer to each other

curtesy of the same. Mannell et al. [64] cite norms and relationships among young people as rapidly changing due to the influence of technology and social media. They cite cellular phone use and social media as having increased substantially across Southern Africa for example, creating new spaces for health interventions and social interaction [69, 70]. The need to bear cognizance of the influence of social media during intervention design and implementation for adolescents and young people can therefore not be overstated.

In line with modern trends youthful stakeholders also highlighted the need in implementing and scaling up the intervention to involve youthful mentors and facilitators with the natural zeal, passion and motivation to serve fellow youth. Further there was mention of the need to rely on already existing youth structures in communities that might just need empowerment and capacity building. The need to foster local platforms for youth in Malawi rather than international ones that only target a few young people like the Barrack Obama or Nelson Mandela Foundations was also mentioned. Above all else the need for youth to focus on the impact of interventions rather than on accruing incentives was another striking observation. All these findings underscore the need for innovation in interventions for adolescents and young people to resonate with youth expectations, priorities and needs to ensure long-term effectiveness, impact and sustainability [53, 64, 71, 72].

## Strengths and limitations of the study

Our study is one among intervention studies with a component involving seeking input from stakeholders in their capacity as providers, designers and evaluators of programmes on the ground. There is often a gap between research, policy and practice and we endeavored to bridge the gap through this stakeholder acceptability study. Our study being qualitative however had the following limitations. The major limitation was the subjectivity associated with qualitative processes of data collection. In essence, stakeholders in organizations might have been constrained by mandates guiding their projects and programmes especially as envisioned by their funders. Our acceptability assessment may also have been based on personal opinions of stakeholders some of which may lack objectivity, may be subjective and perhaps bias findings.

## Conclusion

The study mainly aimed to determine the acceptability and odds of potential for scale up and sustainability of the RRBM model in adolescent sexual behaviour change and HIV risk reduction. Stakeholders had consensus over the acceptability and potential sustainability of the intervention. They also suggested areas that might need modification or improvement. The need to involve traditional leaders, parents and other change agents was recommended. There was also a recommendation for including structural components such as economic empowerment of girls when implementing so as to target distal underlying and root causes of behaviour. Stakeholders also took cognizance of the need for interventions to focus on modern trends driving youth subcultures and sexual dynamics such as social media. Interventions need to be context and population specific and not static to be consistent with global trends as technology continues to bring millennial youth much closer to one another than ever before.

## Supporting information

**S1 File. Risk reduction behavioural model (detailed description).**
(DOCX)

**S2 File. Interview guide.**
(DOCX)

## Acknowledgments

The authors wish to acknowledge the contributions of all the participants involved in this study as well as the organizations and school authorities whose input is invaluable and will go a long way in improving health outcomes for HIV risk reduction among adolescents and young people in Malawi. We also thank College of Medicine for the support they offered. Special thanks to the Research Support Center at College of Medicine as well as the Malawi-Liverpool-Wellcome Trust Clinical Research Programme for the advanced postgraduate research methods courses in qualitative and quantitative research methods as well as data analysis.

## Author Contributions

**Conceptualization:** Marisen Mwale.

**Data curation:** Marisen Mwale.

**Formal analysis:** Marisen Mwale.

**Investigation:** Marisen Mwale.

**Methodology:** Marisen Mwale.

**Project administration:** Adamson S. Muula.

**Validation:** Adamson S. Muula.

**Writing – original draft:** Marisen Mwale.

**Writing – review & editing:** Adamson S. Muula.

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
