## [Decision Letter · Decision Letter 0]

21 Jan 2021

PONE-D-20-17629

Stakeholder acceptability of the risk reduction behavioural model [RRBM] as an alternative model for adolescent HIV risk reduction and sexual behavior change in Northern Malawi

PLOS ONE

Dear Dr. Mwale,

Thank you for submitting your manuscript to PLOS ONE. After careful consideration, we feel that it has merit but does not fully meet PLOS ONE’s publication criteria as it currently stands. Therefore, we invite you to submit a revised version of the manuscript that addresses the points raised during the review process.

We look forward to receiving your revised manuscript.

Kind regards,

Wendee Wechsberg

Academic Editor

PLOS ONE

Journal Requirements:

2. When reporting the results of qualitative research, we suggest consulting the COREQ guidelines: http://intqhc.oxfordjournals.org/content/19/6/349. In this case, please consider including more information on the number of interviewers, their training and characteristics; and whether bias issues were considered. Moreover, please provide the interview guide used.

3.In your Data Availability statement, you have not specified where the minimal data set underlying the results described in your manuscript can be found. PLOS defines a study's minimal data set as the underlying data used to reach the conclusions drawn in the manuscript and any additional data required to replicate the reported study findings in their entirety. All PLOS journals require that the minimal data set be made fully available. For more information about our data policy, please see http://journals.plos.org/plosone/s/data-availability.

4.Thank you for stating the following in the Funding Section of your manuscript:

"The Government of the Republic of Malawi through Mzuzu University funded the study."

 "The author(s) received no specific funding for this work"

5.Your ethics statement should only appear in the Methods section of your manuscript. If your ethics statement is written in any section besides the Methods, please delete it from any other section.

Reviewers' comments:

Reviewer's Responses to Questions

**Comments to the Author**

1. Is the manuscript technically sound, and do the data support the conclusions?

Reviewer #1: Yes

Reviewer #2: Partly

2. Has the statistical analysis been performed appropriately and rigorously? 

Reviewer #1: I Don't Know

Reviewer #2: N/A

3. Have the authors made all data underlying the findings in their manuscript fully available?

Reviewer #1: Yes

Reviewer #2: Yes

4. Is the manuscript presented in an intelligible fashion and written in standard English?

Reviewer #1: Yes

Reviewer #2: Yes

5. Review Comments to the Author

Reviewer #1: interesting study in an important public and reproductive health domain that is how to engage stakeholders in the highly sensitive domain of sexual and reproductive health/HIV in an adolescent population. The methodology and the findings are relevant for Malawi but also for sharing with other countries in the region

Reviewer #2: This qualitative study explored the acceptability, sustainability, and need for modifications of an intervention model. The main strength of this article was that implementers and teachers who work in adolescent sexual and reproductive health were interviewed. There were several areas where the article could be improved. The intervention model in which the implementers were asked to react to was not described thus it was difficult to understand what specific aspects of the interventions were found to be acceptable or sustainable. There were several gaps in the methods sections and the themes described did not align well with the quotes. I offer the following suggestions to strengthen the article.

Introduction

- It is unclear what the RRBM is and how it differs from the RRBI. Describe the components of the intervention, the duration of the intervention, the target population, and the specific behaviors the intervention is meant to address

- Examining triability and feasibility were considered part of the purpose of the study but this was not explored in the methods, results, or discussion. The article would be strengthened by grounding the article in concepts from implementation science. Consider using the following article:

o Proctor, E., Silmere, H., Raghavan, R., Hovmand, P., Aarons, G., Bunger, A., Griffey, R., & Hensley, M. (2011). Outcomes for implementation research: conceptual distinctions, measurement challenges, and research agenda. Administration and policy in mental health, 38(2), 65–76. https://doi.org/10.1007/s10488-010-0319-7

- The section “purposes and objectives” should be cut as it is redundant from the previous paragraph.

Methods

- Page 6: The justification for selecting individual interviews over group interviews/workshops does not need to be included. Cut the following sentences:

o “The choice of the individualized process compared to a stakeholder workshop was considered cost-effective. Above all else individualized interviews allowed input and accorded each stakeholder enough time to appraise the model and intervention comprehensively. In our assessment, this could have been compromised in a workshop scenario due to constraints that arise because of time limitations and perhaps group dynamics.”

- Page 6: Provide more details about the interview process. Where were interviews conducted? Who conducted the interviews? Approximately how long did they last?

- Page 7: Provide more information about the data analysis process. How many people were involved in analysis? How did they reach consensus? Was analysis software used?

Results

- Page 10: The quotes and associated text for theme 2 do not address implementation fidelity. The definition is, “…the degree to which an intervention was implemented as it was prescribed in the original protocol or as it was intended by the program developers (A review of research on fidelity of implementation: implications for drug abuse prevention in school settings. Dusenbury et al. 2003; A glossary for dissemination and implementation research in health. Rabin et al. 2008).” The quotes about the intervention needing to meet the needs of rural and out of school girls and girls with disabilities is closer to the implementation outcome of “reach.” Authors need to describe how teacher conservatism can affect fidelity.

- Page 12: Provide more of a description of “youth priorities and needs.” The quotes included in the table aren’t all aligned with that theme. For example, the quote, “focus in implementation should be on already existing structures in communities but perhaps lacking empowerment and capacity building,” doesn’t seem like a youth priority or need.

- Page 15: The last two quotes in the table more address the theme of sustainability in theme 5.

Discussion

- Page 18: Paragraphs two and three discuss structural and combination intervention approaches, however that was minimally discussed in the results section. Only one quote in table 2 describes the need for structural interventions. Cut those paragraphs.

- Page 21: The first strength that this study is one of the few studies that includes stakeholder involvement can be tempered a little bit. There are many studies that involve stakeholder interviews.

6. PLOS authors have the option to publish the peer review history of their article (what does this mean?). If published, this will include your full peer review and any attached files.

Reviewer #1: No

Reviewer #2: No

---

## [Author Response · Author response to Decision Letter 0]

5 Feb 2021

Dear Editor/(s)

Thank you for the decision for us to revise our article # PONE-D-20-17629 – title, ‘Stakeholder acceptability of the risk reduction behavioural model [RRBM] as an alternative model for adolescent HIV risk reduction and sexual behaviour change in Northern Malawi’.

We have compiled our revisions beginning with those associated with PLOS One requirements and followed by revisions on queries raised by reviewers. In our rebuttal we first outline the query raised by either editor or reviewer and present its associated response as follows:

Revisions – Editorial PLOS One requirements

1. Query – Please ensure that your manuscript meets PLOS ONE’S style requirements including those for file naming:

Response – Formatting of manuscript main body, files, title and author affiliations done in line with PLOS One guidelines.

2. Query - When reporting the results of qualitative research, we suggest consulting the COREQ guideline. Moreover, please provide the interview guide used

Response – COREQ guidelines for reporting qualitative results have been adhered to in line with - [Tong A, Sainsbury P & Craig J. Consolidated criteria for reporting qualitative research (COREQ): a 32-item checklist for interviews and focus groups. International Journal for Quality in Health Care, 2007; 19(6): 349 – 357]. The interview guide has been included as a supporting file – SI File. File 1 Title- Interview guide. 

\\

3. Query - In your Data Availability statement, you have not specified where the minimal data set underlying the results described in your manuscript can be found …..

Response – Minimal data set has been deposited in draft with Qualitative Data Repository (QDR) – University of Syracuse, and can be accessed at https://doi.org.10.5064/F6RGY6T3

4. Query – We note that you have provided funding information that is not currently declared in your Funding Statement. However, funding information should not appear in the Acknowledgements section or other areas of your manuscript. We will only publish funding information present in the Funding section of the online submission form.

Please remove any funding-related text from the manuscript and let us know how you would like to update your Funding Statement.

Please include your amended statement within your cover letter; we will change the online submission form on your behalf.

Response- Funding related text has been removed from any section in the manuscript and has been instead included in the cover letter.

5. Query – Your ethics statement should only appear in the Methods section of your manuscript. If your ethics statement is written in any section besides the Methods, please delete it from any other section.

Response – The ethics statement has been included only in the Methods section and removed from declarations section with the entire declarations section removed.

Revisions – Reviewer comments

Introduction

1. Query - It is unclear what the RRBM is and how it differs from the RRBI. Describe the components of the intervention, the duration of the intervention, the target population, and the specific behaviours the intervention is meant to address.

Response – The RRBM intervention has been presented as a separate sub-section in the methods with title, ‘The RRBM as an alternative model for HIV risk reduction among adolescents’ - (pages: 10-17) 

Components of the intervention as it was tested for efficacy, the duration of the intervention, the target population and the specific behaviours model was meant to address are presented in another sub-section of the methods, title – ‘The RRBM model as tested for efficacy through a peer education intervention with adolescent participants in Northern Malawi’ - (pages: 18-19)

2. Query – Examining triability and feasibility were considered part of the purpose of the study but this was not explored in the methods, results or discussion. The article would be strengthened by grounding the article in concepts from ‘Implementation science’. Consider using Proctor E et al (2011).

Response – As recommended and in line with concepts from ‘Implementation science’ – Proctor et al (2011); the concepts triability and feasibility have been removed from manuscript - (pages: 2 & 3) and the same also applies to replacing the concept fidelity with the concept effectiveness in the results section (page – 22).

Methods

3. Query – Page 6 – The justification for selecting individual interviews over group interviews /workshop does not need to included. Cut the following sentence, ‘The choice of the individualized process compared …….’

Response – The sentence per se has been removed as recommended (page 6).

4. Query – Page 6 – Provide more details about the interview process. Where were interviews conducted? Who conducted the interviews? Approximately how did they last?

Response – Details about the interview process have been presented as recommended; including where interviews were conducted, the interviewer and duration (page 7)

5. Query – Page 7 - Provide more information about the data analysis process. How many people were involved in analysis? How did they reach consensus? Was analysis software used?

Response – As recommended, information about the data analysis process; including number of people involved, how consensus was reached, and whether software was used has been presented on (page 8).

Results

6. Query – Page 10 – The quotes and associated text for Theme 2 do not address implementation fidelity as defined in Rabin et al (2008).

Response – The concept ‘fidelity’ has been replaced by the concept ‘effectiveness’ in associated text for Theme 2 to align with intended objective (page 22 & 32).

7. Query – The quotes about the intervention needing to meet the needs of rural and out of school girls and girls with disabilities is closer to the implementation outcome of ‘reach’. Authors need to describe how teacher conservatism can affect fidelity.

Response – Point was taken and addressed with the replacement of the concept fidelity with the concept effectiveness as highlighted in 6 above.

8. Query – Page 12 – Provide more of description of ‘youth priorities and needs’. The quotes included in the table aren’t all aligned with that theme for example, the quote, ‘focus in implementation should be on already existing structures in communities but perhaps lacking empowerment and capacity building’ doesn’t seem like a youth priority or need.

Response – The quote and other sections have been qualified with the concept youth as intended where necessary to align with the theme and objective and revised as follows:

‘focus in implementation should be on already existing youth structures in communities but perhaps lacking youth empowerment and capacity building’ (page 24 & 25). 

9. Query – The last two quotes in the table more address the theme of sustainability in Table 5.

Response – The quotes per se have been moved to the theme of sustainability in previously Table 5, but now Table 6 (page 29).

Discussion

10. Query – Page 18 – Paragraphs two and three discuss structural and combination intervention approaches, however that was minimally discussed in the results section. Only one quote in Table 2 describes the need for structural interventions. Cut those paragraphs.

Response – The paragraphs on structural and combination interventions have been removed as recommended (page 30 – highlighted).

11. Query – The first strength that this study is one of the few studies that include stakeholder involvement can be tampered a little bit. There are many studies that involve stakeholder interviews.

Response – The statement has been revisited as follows: ‘Our study is one among intervention studies with a component involving seeking input from stakeholders …’ (page 33 - highlighted)

We hope the responses cover all the queries raised in the review.

Regards,

Marisen Mwale

---

## [Decision Letter · Decision Letter 1]

20 Sep 2021

PONE-D-20-17629R1Stakeholder acceptability of the risk reduction behavioural model [RRBM] as an alternative model for adolescent HIV risk reduction and sexual behavior change in Northern MalawiPLOS ONE

Dear Dr. Mwale,

Thank you for submitting your manuscript to PLOS ONE. After careful consideration, we feel that it has merit but does not fully meet PLOS ONE’s publication criteria as it currently stands. Therefore, we invite you to submit a revised version of the manuscript that addresses the points raised during the review process.

 The authors have sufficiently responded to the reviewers comments. However, the section "The RRBM as an alternative model..." is too long and detracts from the overall manuscript. Reviewer #1 has suggested cutting this section. As an alternative, the authors could reduce this to a single paragraph and include the remainder as a supplement. This would significantly improve the manuscript's focus and flow. Please submit your revised manuscript by Nov 04 2021 11:59PM. If you will need more time than this to complete your revisions, please reply to this message or contact the journal office at plosone@plos.org. Please include the following items when submitting your revised manuscript:A rebuttal letter that responds to each point raised by the academic editor and reviewer(s). You should upload this letter as a separate file labeled 'Response to Reviewers'.A marked-up copy of your manuscript that highlights changes made to the original version. You should upload this as a separate file labeled 'Revised Manuscript with Track Changes'.An unmarked version of your revised paper without tracked changes. You should upload this as a separate file labeled 'Manuscript'.If applicable, we recommend that you deposit your laboratory protocols in protocols.io to enhance the reproducibility of your results. Protocols.io assigns your protocol its own identifier (DOI) so that it can be cited independently in the future. For instructions see: https://journals.plos.org/plosone/s/submission-guidelines#loc-laboratory-protocols. Additionally, PLOS ONE offers an option for publishing peer-reviewed Lab Protocol articles, which describe protocols hosted on protocols.io. Read more information on sharing protocols at https://plos.org/protocols?utm_medium=editorial-email&utm_source=authorletters&utm_campaign=protocols.

We look forward to receiving your revised manuscript.

Kind regards,

Brian C. Zanoni, MD

Academic Editor

PLOS ONE

Journal Requirements:

Additional Editor Comments (if provided):

The authors have sufficiently responded to the reviewers comments. However, the section "The RRBM as an alternative model..." is too long and detracts from the overall manuscript. Reviewer #1 has suggested cutting this section. As an alternative, the authors could reduce this to a single paragraph and include the remainder as a supplement. This would significantly improve the manuscript's focus and flow.

Reviewers' comments:

Reviewer's Responses to Questions

**Comments to the Author**

1. If the authors have adequately addressed your comments raised in a previous round of review and you feel that this manuscript is now acceptable for publication, you may indicate that here to bypass the “Comments to the Author” section, enter your conflict of interest statement in the “Confidential to Editor” section, and submit your "Accept" recommendation.

Reviewer #1: All comments have been addressed

Reviewer #2: (No Response)

2. Is the manuscript technically sound, and do the data support the conclusions?

Reviewer #1: Yes

Reviewer #2: Yes

3. Has the statistical analysis been performed appropriately and rigorously? 

Reviewer #1: Yes

Reviewer #2: N/A

4. Have the authors made all data underlying the findings in their manuscript fully available?

Reviewer #1: Yes

Reviewer #2: Yes

5. Is the manuscript presented in an intelligible fashion and written in standard English?

Reviewer #1: Yes

Reviewer #2: Yes

6. Review Comments to the Author

Reviewer #1: I have carefully read the paper as well as the responses to earlier comments and questions that have been well addressed in my view. No additional issues to be raised; interesting paper, well written

Reviewer #2: Thank you for the opportunity to review the revised manuscript. The manuscript is much improved and the majority of my comments have been addressed. One area that needs to be revised is in the description of the RRBM. While it was helpful to read a description, it was quite long. The section titled, "The RRBM model as tested for efficacy through a peer-education

intervention with adolescent participants in Northern Malawi" provides sufficient detail. Rename that section "Risk Reduction Behavioral Model." Cut the section "The RRBM as an alternative model for HIV risk reduction among adolescents" on pages 10-17.

7. PLOS authors have the option to publish the peer review history of their article (what does this mean?). If published, this will include your full peer review and any attached files.

Reviewer #1: **Yes: **Marleen Temmerman, Aga Khan University, Nairobi Kenya

Reviewer #2: No

---

## [Author Response · Author response to Decision Letter 1]

24 Sep 2021

Dear Editor/(s)

Thank you for the decision for us to revise our article # PONE-D-20-17629R1 – title, ‘Stakeholder acceptability of the risk reduction behavioural model [RRBM] as an alternative model for adolescent HIV risk reduction and sexual behaviour change in Northern Malawi’.

We have compiled our revisions beginning with those associated with PLOS One requirements and followed by revisions on queries raised by reviewers. In our rebuttal we first outline the query raised by either editor or reviewer and present its associated response as follows:

Revisions – Editorial PLOS One requirements

1. Query – Please review your references list to ensure that it is complete and correct. If you have cited papers that have been retracted, please include the rationale for doing so in the manuscript text, or remove these references and replace them with relevant current references. Any changes in the reference list should be mentioned in the rebuttal letter that accompanies your revised manuscript.

Response – References have been revised as recommended. Reference 61 has been swapped with reference 26 to suit the link of socio-ecological and socio-cultural models (citations 26, 27 and 28) as adapted in the paper (page 40).

2. Query - The authors have sufficiently responded to the reviewer comments. However, the section ‘The RRBM as an alternative model …’ is too long and detracts from the overall manuscript. Reviewer # 1 has suggested cutting this section. As an alternative, the authors could reduce this to a single paragraph and include the remainder as a supplement. This would significantly improve the manuscript focus and flow.

Response – As recommended by the reviewer, the section ‘The RRBM as an alternative model ….’ Has been cut and the content has been retained as an additional supplementary file 1 (S1 File 1. Title – The Risk reduction behavioural model).

Revisions – Reviewer comments

1. Query – The manuscript is much improved and the majority of my comments have been addressed. One area that needs to be revised is in the description of the RRBM. While it was helpful to read a description, it was quiet long. The section titled, ‘The RRBM model as tested for efficacy through a peer-education intervention with adolescent participants in Northern Malawi’ provides sufficient detail. Rename that section ‘The Risk Reduction Behavioural Model.’ Cut the section ‘The RRBM as an alternative model for HIV risk reduction among adolescents’.

Response – As recommended by Reviewer # 2, the section titled ‘The RRBM model as tested for efficacy through a peer-education intervention with adolescent participants in Northern Malawi’ has been renamed to ‘The Risk Reduction Behavioural Model’ with a bit of revisions (pages 10 - 11). 

 Additionally the section ‘The RRBM as an alternative model for HIV risk reduction among adolescents’ has been cut as recommended and the content thereof included as a detailed description of the risk reduction model in supplementary file 1 (S1 File 1. Title – The risk reduction behavioural model).

We hope the responses cover all the queries raised in the review.

Regards,

Marisen Mwale

---

## [Editor Report · Decision Letter 2]

30 Sep 2021

Stakeholder acceptability of the risk reduction behavioural model [RRBM] as an alternative model for adolescent HIV risk reduction and sexual behavior change in Northern Malawi

PONE-D-20-17629R2

Dear Dr. Mwale,

We’re pleased to inform you that your manuscript has been judged scientifically suitable for publication and will be formally accepted for publication once it meets all outstanding technical requirements.

Kind regards,

Brian C. Zanoni, MD

Academic Editor

PLOS ONE

Additional Editor Comments (optional):

The revised manuscript is suitable for publications.
---

## [Editor Report · Acceptance letter]

8 Oct 2021

PONE-D-20-17629R2 

Stakeholder acceptability of the risk reduction behavioural model [RRBM] as an alternative model for adolescent HIV risk reduction and sexual behavior change in Northern Malawi 

Dear Dr. Mwale:

I'm pleased to inform you that your manuscript has been deemed suitable for publication in PLOS ONE. Congratulations! Your manuscript is now with our production department. 

Kind regards, 

on behalf of

Dr. Brian C. Zanoni 

Academic Editor

PLOS ONE